

# Hole in One: an element reduction approach to modeling bone porosity in finite element analysis

Beatriz L. Santaella[1] and Z. Jack Tseng[1,2,3]

[1] Department of Pathology and Anatomical Sciences, Jacobs School of Medicine and Biomedical Sciences, State University of New York, Buffalo, NY, United States of America
[2] Department of Integrative Biology and Museum of Paleontology, University of California, Berkeley, CA, United States of America
[3] Division of Paleontology, American Museum of Natural History, New York, NY, United States of America

## ABSTRACT

Finite element analysis has been an increasingly widely applied biomechanical modeling method in many different science and engineering fields over the last decade. In the biological sciences, there are many examples of FEA in areas such as paleontology and functional morphology. Despite this common use, the modeling of trabecular bone remains a key issue because their highly complex and porous geometries are difficult to replicate in the solid mesh format required for many simulations. A common practice is to assign uniform model material properties to whole or portions of models that represent trabecular bone. In this study we aimed to demonstrate that a physical, element reduction approach constitutes a valid protocol for addressing this problem in addition to the wholesale mathematical approach. We tested a customized script for element reduction modeling on five exemplar trabecular geometry models of carnivoran temporomandibular joints, and compared stress and strain energy results of both physical and mathematical trabecular modeling to models incorporating actual trabecular geometry. Simulation results indicate that that the physical, element reduction approach generally outperformed the mathematical approach: physical changes in the internal structure of experimental cylindrical models had a major influence on the recorded stress values throughout the model, and more closely approximates values obtained in models containing actual trabecular geometry than solid models with modified trabecular material properties. In models with both physical and mathematical adjustments for bone porosity, the physical changes exhibit more weight than material properties changes in approximating values of control models. Therefore, we conclude that maintaining or mimicking the internal porosity of a trabecular structure is a more effective method of approximating trabecular bone behavior in finite element models than modifying material properties.

# INTRODUCTION

Finite element analysis (FEA) is a continuum mechanics-based technique originally conceived and used in the engineering design process to predict the behavior (i.e., response)

Corresponding author
Beatriz L. Santaella,
bsantaella@outlook.com

of structures to prescribed loading conditions. This technique uses discretized representations of real-world structures, thereby enabling the design of these systems to be optimized mathematically with minimum physical prototyping and testing (*Dumont, Grosse & Slater, 2009*; *Zienkiewicz & Taylor, 2000*). With advances in computer software packages that allow a seamless connection of FEA to CAD and image data based modeling, the simulation method has also been applied to functional morphological research in organismal biology, including extinct organisms (as reviewed in *Ross, 2005*; *Rayfield, 2007*; *Tseng & Wang, 2010*; *Bright, 2014*). FEA of feeding mechanics of living and extinct vertebrates have been used in comparative functional morphology for more than a decade (*Rayfield, 2005*; *Alexander, 2006*; *Barrett & Rayfield, 2006*; *McHenry et al., 2006*; *Thomassen et al., 2007*), and the method also has been applied in studies in other organismal systems such as insect flight and mechanoreception (*Combes & Daniel, 2003*; *Dechant et al., 2006*; *Wootton, 2003*), and plant biomechanics (*Fourcaud & Lac, 2003*; *Niklas, 1999*).

For the last decade or so, the boundaries of FEA have been pushed towards more accurate modeling of bone structures to better understand skeletal form and function (*Rayfield, 2007*; *Bourke et al., 2008*; *Wroe et al., 2008*; *Strait et al., 2010*). Still, porous structures like trabecular bone and other complex biological geometries remain problematic in FE modeling given their internal complexity, and the conversion from 2D to 3D of intricate structures that frequently generate errors in elemental overlaps and highly skewed elemental shapes in small anatomical regions. Based on our experience working with bone meshes, biological structures with a high amount of trabecular bone or porous components have higher chances of meshing errors in the FE solid meshing process (but see *Fagan et al., 2007* for an alternative, albeit computationally more intensive, and resolution limiting, voxel-based modeling approach). When modeling this type of porous structure, it is common to avoid the complexity of creating a detailed trabecular network by modeling entire models as homogeneous cortical bone and ignoring trabecular geometry, and/or changing the material properties in different element groups within a model to represent cortical versus trabecular bones (*Strait et al., 2005*; *Strait et al., 2009*; *Wroe, 2008*; *Attard et al., 2011*; *Chamoli & Wroe, 2011*). This general simplification approach is used in most comparative studies using FEA that incorporate trabecular morphology, even though it has been demonstrated that trabecular structures can play a very important role in the performance of a mesh when using FEA (*Parr et al., 2013*).

Our objective in this study is to test an alternative, mechanical approach to trabecular bone modeling as a viable solution in addition to mathematical approaches (i.e., changing the material properties of solid models). Potential solutions to accommodate trabecular morphology in finite element modeling that can bypass time-consuming and scan resolution-dependent micro-modeling of trabecular structures are desired. We aim to test the hypothesis that percentage porosity adjustments in solid finite element meshes will generate simulation results comparable or closer to those using actual trabecular morphology, compared to solid models using only modified material property parameter values to simulate trabecular bone behavior.

## MATERIALS AND METHODS

We used five species samples to test a finite element reduction approach to trabecular bone modeling relative to actual trabecular structural models. Each species-specific test sample is represented by three types of experimental cylindrical models: one control cylinder ("CC"); one physically modified cylinder ("PC"); and one material-modified cylinder ("MC"). Definitions of each cylinder model are given below.

### Control group cylinders

The spongy bone cylinder core meshes were taken from *Wysocki & Tseng (2018)*, based on scans of carnivoran (Carnivora, Mammalia) skull specimens from the American Museum of Natural History (*Arctonyx collaris*; *Bassariscus astutus*; *Enhydra lutris*; *Mellivora capensis*; *Vulpes vulpes*) (see Table S1 for scanning parameters). We emphasize that this is not a full-scale comparative analysis; the species were selected based on the relative fill volume range (the amount of space within a predefined digital cylinder sample of trabecular network within the temporomandibular joints of each species that is space versus those that are bone; *Wysocki & Tseng, 2018*). Our sample choices allowed testing of each trabecular material modeling method over a relatively wide range of naturally occurring variations in trabecular density. The range of relative fill volumes span from 7.8% porosity (or 92.2% bone) in *Mellivora capensis* to 46.6% porosity (or 53.4% bone) in *Bassariscus astutus*. These specimen-derived cylinders correspond to a control group to serve as a reference for PC and MC model approximations of von Misses stress and total strain energy. Von Mises stress is a good predictor of failure under ductile fracture, and an appropriate metric for comparing the relative strength of models of bones; strain energy is a measure of the work done in deforming a structure, and is a metric of the degree of overall stiffness or degree of deformation of a structure (*Dumont, Grosse & Slater, 2009*).

Full cylinders corresponding to the maximum, solid volumes possible for the virtual cylindrical cores used in *Wysocki & Tseng (2018)* were designed in Geomagic Wrap 2017.0.1.19 (3D Systems, Rock Hill, South Carolina) with a 10 mm height and 5 mm diameter. Ten cylinders were created, five to be modified by physical element reduction to increase porosity, and the other five to be modified in their material properties but not physical geometry (i.e., they remain solid cylinders). When finished, the cylinders were exported as binary stereolithographic files (.stl). These models serve as input for further processing in the finite element simulation software.

### Materially modified cylinder group

We defined the material properties to apply in all the meshes in the CC and PC experimental groups based on values used in numerous previous FEA studies (*Cowin, 1989*; *Erickson, Catanese III & Keaveny, 2002*), a Young's Modulus of 20 GPa and a Poisson's Ratio of 0.3. For the MC group, the Young's Modulus is adjusted within a range (from 7 GPa to 22 GPa) that is linearly proportional to the density values of the control cylinder (actual species trabecular geometry) for that experimental group's relative fill volume. Relative fill volume ($mm^3$) was calculated using the species-derived 3D model that served as the standard (*Wysocki & Tseng, 2018*). The range was kept in between 7 GPa to 22 GPa in Young's

Modulus to fully encompass the range observed for cortical bone in the literature (*Cowin, 1989*; *Erickson, Catanese III & Keaveny, 2002*). A high porosity physically modified cylinder model (i.e., 46.6% porosity, with 53.4% of the volume being solid) will be associated with a low Young's Modulus (i.e., 7 GPa) materially modified cylinder model in our study. The remaining boundary conditions for the MC group were set up as in the CC group (for a detailed description see Model Simulation Parameters).

## Physically modified cylinder group

A set of the solid meshed cylinders were post-processed using a custom script built in R 3.5.1 (R Foundation for Statistical Computing, Vienna, Austria) that created an induced porosity into cylinder models by randomized solid element removal (full script is available at https://github.com/BeaSantaella/Hole-in-One.git). After importing a solid mesh file from Strand7 into R, then designating a user-defined amount of tetrahedral deletion (as a percentage), the script goes through all the brick elements (which form the structure modeled, and are formed by individual, four-noded tetrahedral elements) and randomly removes the designated percentage of elements from the model. Each tetrahedral element can be randomly selected for removal only once; in other words, randomized selection of elements for removal is done without replacement. The script output is a text file (.txt) in Strand7 format, which can be read back into the simulation software for further analysis.

Each script was assigned a percentage of material deletion based on the relative fill volume of their corresponding control group attributes (26.1% for *Arctonyx collaris*; 46.6% for *Bassariscus astutus*; 16.5% for *Enhydra lutris*; 7.8% for *Mellivora capensis*; 35.8% for *Vulpes vulpes*).

## Script analyses: does random element reduction deliver consistent results?

Prior to comparing PC models to the CC group or MC group, we tested an additional set of 5 models to ascertain the internal consistency of the script (whether random element deletion delivers consistent results).

We applied the same script, set at 16.5% volume deletion (we chose 16.5% deletion as a mid-range value through our tested range), to five otherwise identical solid cylinder models. The remaining parameter values, such as material properties (Young's Modulus: 20 GPa and Poisson's Ratio: 0.3), the amount of force applied (1,000 N), nodes retrained (four nodes, at the end of a cross-section, at the bottom of the cylinder), and the area of application all remained identical (see Model Simulation Parameters). All the points sampled were identical through all of the five cylinders (Fig. 1).

If large differences in magnitude of the stress values are present in script-generated models across different replicates, the script would not represent a true randomized approach to element reduction. If the effects of the script are random, the variability in the results for all 5 additional models should be within comparable ranges of variation. Some variability is expected because the script is based on a random pattern. As a consequence, some arbitrary associations that affect stress values may occur. Overall, our assumption is
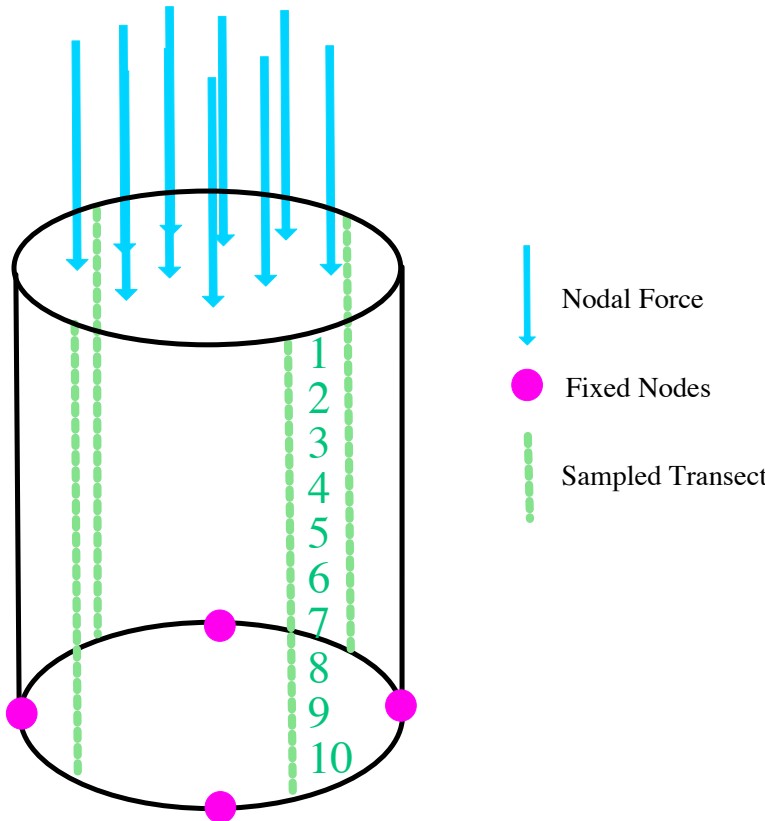

**Figure 1  Locations of boundary conditions on the cylinder models (fixed nodes; force and sample transect).**

that replication of porosity in trabecular structures by random reduction of solid element would result in replication of overall trabecular mechanical behavior.

## Combined physically modified and materially modified cylinders

In order to assess the joint efficacy of introducing both physical porosity and modification of material property parameters, another set of models was created. They present the same percentage of deletion to corresponding PC models, but their material properties were also adjusted to reflect those of their corresponding MC models. The PC+MC analysis on Enhydra group (it would have corresponded to 16.5% + 20 GPa) was not conducted as PC+MC since the Physically modified cylinder for 16.5% already had the same material properties (20 GPa is the default setting). We did this in order to avoid having to lines overlapping in the results.

## Model simulation parameters

We use Finite Element Analysis (FEA) software Strand7 2.4.6 (G1D Computing Pty, Sydney, Australia) to solid mesh the surface cylinder models generated in Geomagic Wrap. In FEA, the geometry of the structural system of interest is approximated by a mesh of simple polyhedral shapes called 'finite elements', connected together at 'nodes', which

are the vertices of polyhedra (*Dumont, Grosse & Slater, 2009*). These polyhedra (known as "bricks" in Strand7) constitute a solid mesh filled inward from the triangular faces of the surface mesh that encompass space representing bone (surface meshes were generated in Geomagic Wrap). A mesh formed by bricks is considered a solid mesh, the mesh type used for finite element analysis in the majority of 3D comparative functional morphology studies.

We applied an arbitrary, 1,000 N of force over the nodes on the entire top surface of all cylinder models and recorded nodal stress values (von Mises stress) at four transects in each model. We sampled a total 40 points along the surface of the cylinders (from top to bottom, 10 sampling points per transect). The mean stress values calculated from these nodal transects are used to compare the CC, PC, MC, and PC+MC experimental groups (Fig. 1). The material properties assigned were: Young's Modulus: 20 GPa and Poisson's Ratio: 0.3; and nodes retrained: four nodes, at the end of a cross-section, at the bottom of the cylinder.

For all model categories, total stored strain energy values are also extracted from each simulation run, in order to characterize the overall stiffness or deformation experienced by the models. All analyses were linear static, which means the relationship between the load and the response is linear; and the applied load does not vary with time. Model files for all analyses conducted are available for download at Zendodo (https://doi.org/10.5281/zenodo.3344501).

## RESULTS

Our results show that physically modified cylinder replicates, assigned the same specific settings, have largely uniform outputs (Fig. 2, Tables S2–S3). There was only a small problematic region, located at the bottom (points 8 to 10) of one of the transects in cylinder IV (Fig. 2A, Table S2). Because there are no differences between the cylinders beside the random arrangements that the script may have produced, the higher stress values on the nodes correspond to a more localized deletion at the sampled area. The higher deletion around that area would affect how the applied force is transmitted and distributed in that location, and thereby extend influence to contiguous areas (as subsequent points show higher stress values). This inconsistency effect is diluted by average stress values across analogous locations of the four sampling transects on each cylinder (Fig. 2B, Table S3). Therefore, all subsequent simulated stress values are reported as mean values.

There is a better overall performance of the physically modified cylinders in comparison with materially modified cylinders when referring to the control cylinders. In the experimental groups for 26.1% and 46.6% porosity (Figs. 3C and 3E, Tables S6–S8), we see a consistent performance of the PC. We can see a slightly more accurate overall trend in physically modified cylinders (it underestimates in certain regions and it is not able to imitate the peaks of CC, but replicates the general trend). The particular, the bottom section (sampled nodes 8 to 10) of the PC cylinders has a more accurate performance (relative to the control) than the materially modified cylinders. MCs in both 26.1% and 46.6% models exhibit a linear transect trend with relatively low stress changes.

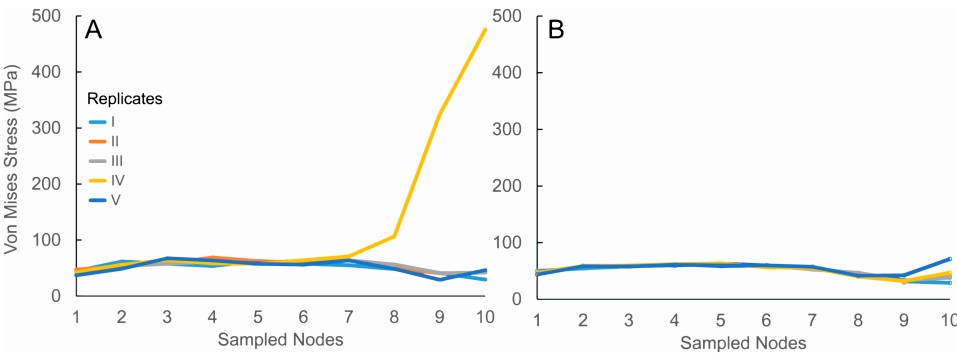

**Figure 2** **Script analyses.** Sampled nodes represent 10 equidistant points along data transects where von Mises stress values were recorded (as described in Table S2). The different cylinder model replicates are labeled from I to V. (A) Stress values obtained from a single transect per replicate. (B) Mean stress values obtained from four transects taken per replicate. Note the presence of aberrant stress values in replicate IV when sampling a single transect that is averaged out in the four-transect sampling approach.

On the other hand, in the experimental group of 16.5% porosity (Fig. 3B, Table S5), PC seemed to be unable to correctly replicate both trend and stress values of the control group. For the experimental group of 7.8% porosity (Fig. 3A, Table S4), PC and MC seem to perform equally well in most of the sampling points (same stress values or off by less than 10 MPa). Except at the beginning and the end (where higher variability may be present, close to the area of force application and nodal restraints), there are minimal differences in stress values across all modeling approaches tested at the lowest porosity level.

In the experimental group of 35.8% porosity (Fig. 3D, Table S7), the differences in stress values seem to be consistent with what we observe in groups with 26.1% and 46.6% porosity (Figs. 3C and 3E, Tables S6–S8). PC replicates the overall CC trend but its values are offset by 60 to 80 MPa, especially at the center core region. MC shows a less accurate trend, with a more linear pattern, and even less resemblance to the CC trend. As seen in all experimental groups (Figs. 3A–3E, Tables S4–S8) the combined PC+MC approach presents the same stress values as the PC group results. The differences are indistinguishable between PC and PC+MC results. The overall stress trends relative to porosity changed are plotted separately for the control versus element-reduction models in Fig. 4.

Strain energy values, used here as an overall measure of the stiffness or degree of deformation experienced by the different models, show that when porosity is low, modifying material properties of solid models provides the closest approximation to trabecular geometry models (Fig. 5; Table 1). However, as porosity increases to 16.5% and above, the element reduction models provide the closest approximation to trabecular geometry model strain energy values. Furthermore, a combined modeling approach of element reduction plus material modification generates close approximations of strain energy values at lower porosities, but is less accurate than either material or element reduction approaches alone, in models with higher porosities (Fig. 5).

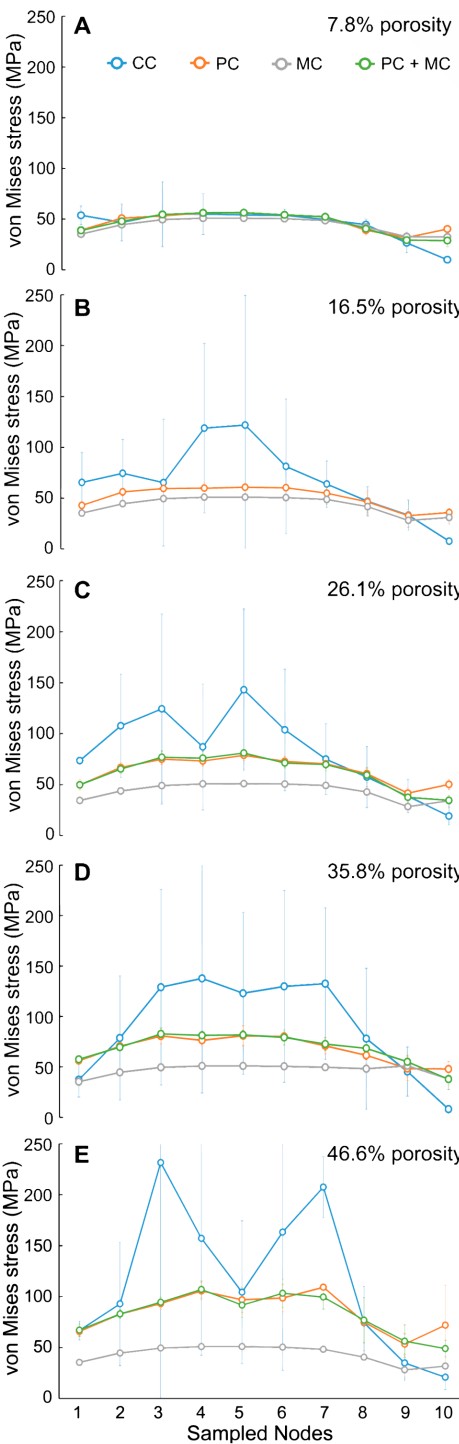

**Figure 3** **Experimental groups 1 to 5.** On the *x*-axis, we display 10 points used to collect the data (point 1 top, point 10 bottom). On the *y*-axis, we show von Mises stress values. Each line is the mean of the four transects sampled (See Tables S4 to S8). The blue line corresponds with the CC; the orange line corresponds with the PC; the grey line corresponds with MC; the green line corresponds with PC+MC. (A) (CC: *Mellivora*; PC: 7.8%; MC: 22GPa), (B) (CC: *Enhydra*; PC: 16.5%; MC: 20GPa), (C) (CC: *Arctonyx*; PC: 26.1%; MC: 16GPa), (D) (CC: *Vulpes*; PC: 35.8%; MC: 10GPa), (E) (CC: *Bassariscus*; PC: 46.6%; MC: 7GPa). Error bars represent 95% confidence intervals around the mean values (See Tables S4 to S8).

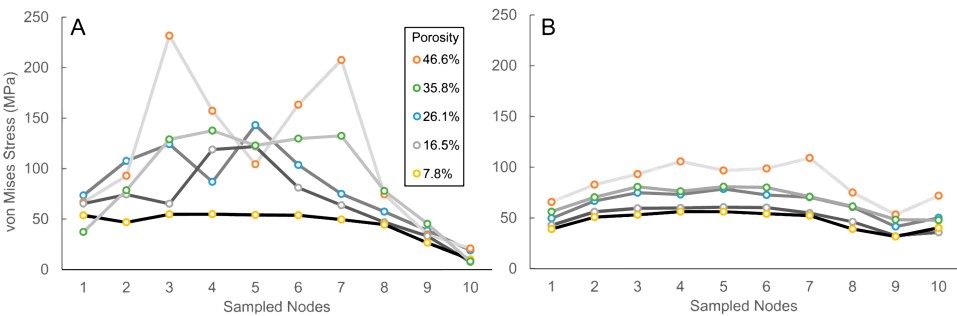

**Figure 4 Von Mises stress values across cylinders, organized by.** (A) control group results and (B) element-reduced group results. Degree of porosity is represented by shading of plot lines (darker shade equals lower porosit, lighter shade equals higher porosity).

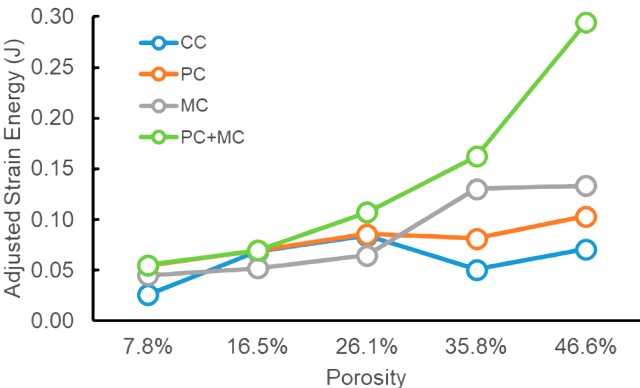

**Figure 5 Adjusted strain energy comparisons for experimental models.** CC, control group cylinder models; PC, physical element reduction models; MC, material property modified models; J, Joules. For numerical values see Table 1.

## DISCUSSION

Element reduction is potentially a more accurate approach for modeling trabecular stress and strain energy than modification of regional material properties. We tested the hypothesis that, even if they are not 100% replicates of trabecular bone models, porous FE models can at least behave in a comparable way, and provide a closer approximation of mechanical behavior than only modifying overall material property parameters of solid models. Our results indicate that an element reduction approach to modeling bone porosity produced stress magnitudes that are generally closer to values generated from models containing actual trabecular bone geometry, compared to only modifying material properties to simulate bone porosity (Figs. 3 and 4). Furthermore, element reduction produces models with strain energy values that are comparable to those estimated by trabecular bone and material property modified models at low porosity values, and values that best approximate trabecular bone model outcomes at higher porosity values out of all the modeling approaches tested (Fig. 5).

**Table 1  Strain energy measurements of experimental models.** Raw strain energy values were adjusted by model volume according to the recommendations of *Dumont, Grosse & Slater (2009)*.

| Porosity | Experiment | SE (J) | Volume (mm3) | Adjusted SE (J) |
|---|---|---|---|---|
| 7.80% | CC | 0.0253 | 180.60 | 0.0253 |
| | PC | 0.0544 | 180.14 | 0.0544 |
| | MC | 0.0436 | 195.79 | 0.0447 |
| | PC+MC | 0.0547 | 180.14 | 0.0547 |
| 16.50% | CC | 0.0703 | 163.51 | 0.0680 |
| | PC | 0.0714 | 164.07 | 0.0692 |
| | MC | 0.0503 | 195.79 | 0.0516 |
| | PC+MC | 0.0714 | 164.07 | 0.0692 |
| 26.10% | CC | 0.0902 | 144.80 | 0.0838 |
| | PC | 0.0921 | 144.05 | 0.0855 |
| | MC | 0.0629 | 195.79 | 0.0646 |
| | PC+MC | 0.1151 | 144.05 | 0.1068 |
| 35.80% | CC | 0.0567 | 125.79 | 0.0503 |
| | PC | 0.0913 | 125.85 | 0.0810 |
| | MC | 0.1266 | 195.79 | 0.1300 |
| | PC+MC | 0.1826 | 125.85 | 0.1620 |
| 46.60% | CC | 0.0845 | 104.71 | 0.0706 |
| | PC | 0.1233 | 104.94 | 0.1031 |
| | MC | 0.1298 | 195.79 | 0.1333 |
| | PC+MC | 0.3522 | 104.94 | 0.2945 |

**Notes.**

CC, control group cylinder models; PC, physical element reduction models; MC, material property modified models; SE, strain energy; J, Joules.

Overall, the peaks and valleys of stress in the trabecular geometry models are not well-replicated by alternative modeling approaches. Stress peaks in the transect plot for the control group might be explained by how close the sampled node was to a physical hole or opening on the model surface (in other words, adjacent to an internal porous network) (Fig. 4A). The nodal values may be influenced by elevated stress values associated with such porosity. Thus, creating a cover layer of plate elements, then sampling from that surface, could be one modeling solution to account for the source of that possible noise. This could be considered in further studies, but our goal for this first study was to compare relative performances between the mechanical approach and the mathematical approach (PC vs MC); rather than specifically creating a protocol to mimic actual bone. As apparent from the results, the overall stress magnitude changes, but not the precise stress peaks, are replicated using the element-reduction approach proposed in this study (Fig. 4B).

It is remarkable that even without a cover of cortical bone (or a thick layer that might homogenize the values at the nodal transect regions) the mechanical modeling approach still has a certain consistency (results are similar in all four experimental groups for PC+MC models). Based on our results, the ability of PC models to approximate stress values in the control group models is best in moderate density models. As shown in Fig. 3D, the overall curvature of the stress sampling transect in the control model are mimicked by PC,
whereas materially modified cylinder trends show a low-sensitivity trajectory, indicating that the overall performance of materially modified cylinders is less accurate than observed for data in the PC group.

We note that the element reduction script generated models with holes in a random pattern, whereas the actual species trabecular geometries contain holes surrounding a network of bony struts. As a consequence, PC models are more homogeneous in how they distribute forces. In other words, when compared to the CC group, the PC models perform as a more rigid material. This is probably related to their lack of internal heterogeneity in arrangements or concentration of large pores/bony struts that may not be represented by the mechanical modeling approach. This is another key factor to consider in future research into improving accuracy of trabecular bone modeling in FE simulations.

It is also quite clear that material properties modified cylinders behave as an even more rigid material than the other two groups. The von Mises stress values, which reflect the likeliness of a certain structure to fail, are apparently lower in MC. This stiffness, or lack of it, may be related to the internal network influence on the overall performance (*Parr et al., 2013*). Our results point to a plausible explanation to why so many published FE analyses report high stiffness values compared to those known for actual bone: most modeling approaches use a "solid approximation" of trabecular bone, but differences in material distribution, which clearly influences stress estimates as per results in this study, are rarely taken into account.

Bone tissue can behave as a homogeneous material on a microscale (*Müller, 2009*) with both individual trabeculae and compact bone having similar material properties (*Rho, Ashman & Turner, 1993*). Therefore, changing material properties to differentiate compact versus trabecular bone may not adequately replicate bone behavior in FE simulations. The adjustment of bone porosity based on the internal density of corresponding trabecular geometry models did better replicating the stress and strain energy values of the control group than MC models (Figs. 3 and 5). Accordingly, overall, the change in the material properties is a less effective way to approximate model mechanical behavior than physically reducing the element density of solid mesh models via the randomization approach (Fig. 6).

In addition, models with both physically introduced porosity and material property changes combined behaved similarly to the models with only introduced porosity, suggesting the dominant role of element reduction in dictating surface mechanical behavior of the cylinder models in our study. This finding could be even more important when considering animals with different amounts of trabecular volume. When modeling low trabecular volumes, such as those in the bones of birds or pterosaurs, the considerations should differ from modeling animals with higher bone densities such as some mammals. Consequently, comparative analysis or paleontological reconstructions should consider the nature of the structure that is being analyzed. An adequate adjustment relative to cortical and trabecular bone ratios when modeling could produce more accurate models that reflect mechanical behavior of trabecular geometries.

All things considered together, we summarize an important take home message for modeling complex geometries of skeletal elements in general: without element reduction or specific modeling of trabecular morphology, comparative FEA studies that include taxa

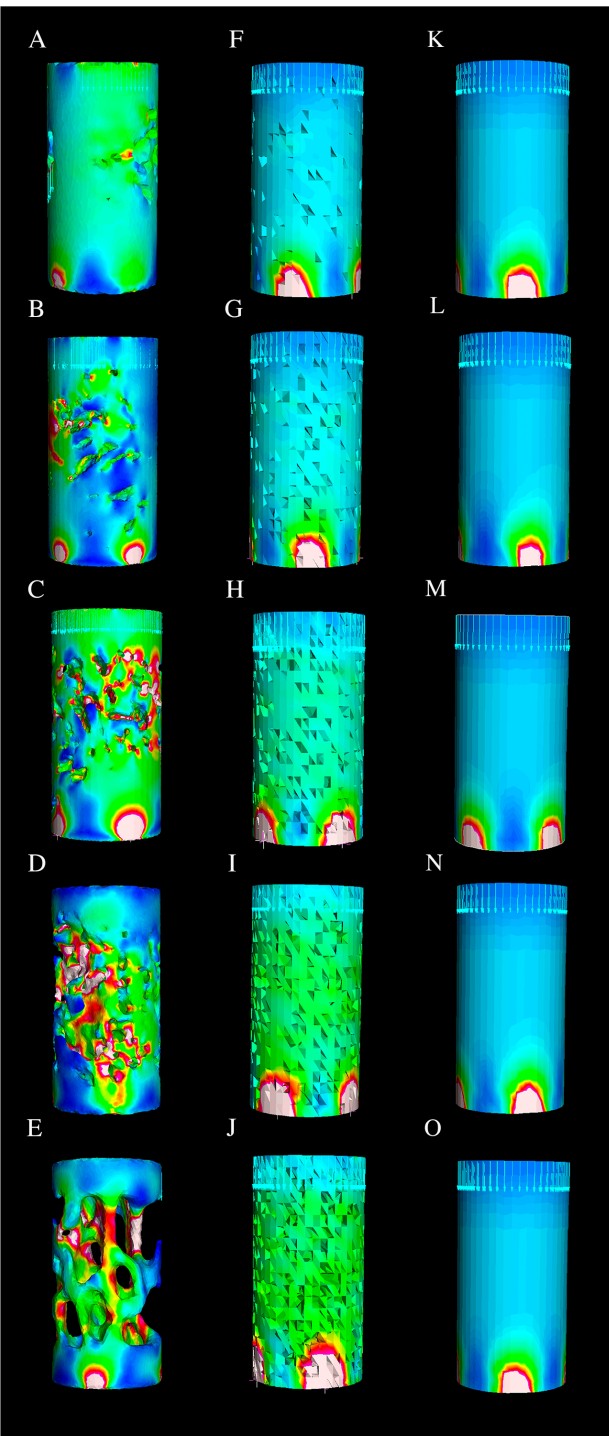

**Figure 6 Visualization of von Mises stress in the cylinders.** Vertically the image is separated into three sections: CC ((A) *Mellivora*, (B) *Enhydra*, (C) *Arctonyx*, (D) *Vulpes*, (E) *Bassariscus*); PC (F. 7.8%, G. 16.5%, H. 26.1%, I. 35.8%, J. 46.6%), and MC (K. 22 GPa, L. 20 GPa, M. 18 GPa, N. 10 GPa, O. 7 GPa).

with a wide range of cortical-to-trabecular bone ratios (e.g., relatively low cortical ratio in vertebrates such as birds and extinct taxa such as pterosaurs) will exhibit larger deviations in both stress and strain energy values than those models that represent morphologies with high cortical ratios. This is an additional source of uncertainty in the modeling protocol that could strongly influence simulation outcomes and thereby, functional morphological inferences made from those simulation results. If close approximation of both stress and strain energy values in finite element simulations involving trabecular structures are of interest, findings from this study demonstrate that an element reduction approach would be preferred over existing material property modification protocols.

## CONCLUSIONS

We demonstrated that an element reduction approach to modeling trabecular structure could more closely simulate behavior of trabecular geometry compared to changing material properties in solid models. Additionally, when element reduction and material property modification methods are combined, the effects of element reduction (i.e., generation of porosity) far outweighs the relative effects of material property modification. We suggest that, unless the complex geometry of trabecular bone is precisely accounted for during the model building process, researchers should first consider modeling the porosity of the material instead of changing material properties. This recommendation is supported by our findings that indicate physical internal porosity generation better approximates mechanical performance (stress and strain energy values) of trabecular structures compared to material property changes. Therefore, we recommend considering bone porosity in such a physical manner in biomechanical modeling of complex trabecular bone geometries in comparative functional morphological studies, as a fast and effective way to approximate trabecular geometry, and to alleviate potential biases in finite element modeling protocol towards taxa that exhibit high trabecular bone ratios in their morphology.

## ACKNOWLEDGEMENTS

We thank J. Flynn, E. Westwig, and M. Chase for providing access to specimens and scanning facility at the AMNH. M. Wysocki provided cylindrical models of the carnivoran species tested in this study. B. Santaella thanks committee members J. Liu and S. Doyle for their time and advice. E. Snively, an anonymous reviewer, and editor P. Cox provided highly constructive comments that significantly improved the visual and textual presentation of this manuscript.

### Funding

Beatriz L. Santaella was funded by a research scholarship from the Functional Anatomy and Vertebrate Evolution Laboratory. The funders had no role in study design, data collection and analysis, decision to publish, or preparation of the manuscript.

### Grant Disclosures
The following grant information was disclosed by the authors:
Functional Anatomy and Vertebrate Evolution Laboratory.

### Competing Interests
The authors declare that there are no competing interests.

### Author Contributions
- Beatriz L. Santaella conceived and designed the experiments, performed the experiments, analyzed the data, prepared figures and/or tables, authored or reviewed drafts of the paper, approved the final draft.
- Z. Jack Tseng conceived and designed the experiments, analyzed the data, contributed reagents/materials/analysis tools, prepared figures and/or tables, authored or reviewed drafts of the paper, approved the final draft.

### Data Availability
Data is available at GITHUB, as BeaSantaella/Hole-in-One https://github.com/BeaSantaella/Hole-in-One.

Data is available at Zenodo: Beatriz Santaella Luna, & Z. Jack Tseng. (2019, July 19). Hole in One. Zenodo. http://doi.org/10.5281/zenodo.3344501

### Supplemental Information
Supplemental information for this article can be found online at http://dx.doi.org/10.7717/peerj.8112#supplemental-information.

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
