# Peer review of "Hole in One: an element reduction approach to modeling bone porosity in finite element analysis"

_PeerJ, doi:10.7717/peerj.8112_

## Round 0.1 · original submission · Minor Revisions

Thanks for submitting your manuscript to PeerJ - please find attached the reviewer comments. Both reviewers are very positive about your study and I agree with them that it represents an exciting potential new method for dealing with trabecular bone in FE models. There are just a few issues that need amending. In particular, can I draw your attention to the following:

1. I agree with reviewer 1 that it is somewhat of an over-interpretation to say that the PC models replicate the peaks and troughs of the CC models. Figure 3 simply doesn't support that. I think you need to tone down the interpretation of the results in this regard. Also, reviewer 1's suggestion of taking some kind of overall or average strain measure to compare models is a very good one I think.

2. Line 270 - you say that VM stresses are 'significantly' lower in MC - did you actually test this statistically?

3. Both reviewers mention clarity of figures, particularly figure 3. The line weights and font sizes definitely need increasing here to improve legibility.

4. Personally, I found the supplementary tables baffling. You have 21 tables, 19 of which have identical titles and legends. Each individual table is quite small - I'm sure these could all be combined into a single file. This would also have the advantage of making it easier to compare between tables. I think each table needs better labelling to indicate which model it is referring to - stick to the CC, PC, MC coding that you've introduced in the manuscript. Also, I'm not sure I would term what you've presented as 'statistical analysis' - it's more 'descriptive statistics'.

Once you've attended to these and the other reviewer comments, I will look forward to seeing a revised version of the manuscript.

Reviewer 1 ·

Basic reporting

Generally fine, see specific comments below

Experimental design

I think some of the methods need slightly more detail to improve clarity, further comments below

Validity of the findings

The findings appear valid, but I think the study would benefit from using an alternative method of assessing the differences between models that is less sensitive to localized variation

Additional comments

This paper compares finite element models of cylinders of real mammalian trabecular bone with models of solid cylinders that have either had their stiffness (material) properties, porosity (physical properties), or both adjusted. The aim is to determine how these modeling choices impact the FE results (i.e. the study is a finite element sensitivity test).
Overall I think this is an interesting study that makes a good contribution to the field. Although the element reduction methods do not mimic the fabric of real bone, the results nevertheless demonstrate the material distribution has a more important effect than material properties when modeling trabecular bone. It is nice to see this quantified, especially as it runs somewhat counter to some of the largely untested conventional wisdom that’s been talked about in FE circles recently. Whether these differences are important enough to make the models more, or less valid, is not addressed, but that is not the point of this study.
I think the paper would benefit from increasingly clarity in some of the methods, and from an enhanced discussion of some of the points I raise above (and below in more detail).

Main Comments
L132: “We defined the material properties to apply in all the meshes in the CC and PC experimental groups (Young’s Modulus: 20 GPa and Poisson's Ratio: 0.3). For the MC group, the Young’s Modulus is adjusted within a range (from 7 GPa to 22 GPa) that is linearly proportional to the density values of the control cylinder (actual species trabecular geometry) for that experimental group’s relative fill volume. Relative fill volume (mm3) was calculated using the species-derived 3D model that served as the standard (Wysocki and Tseng., 2018). The remaining boundary conditions for the MC group were set up as in the CC group.”
I think this needs more explanation. If I’m following this right, each CC has a percentage volume of bone, and this was used to adjust Young’s modulus for the corresponding MC? It took me a lot of re-reads, and the caption for Fig. 3, to wrap my head around this and I’m still not completely sure I get it. A worked example or a figure might be helpful, e.g. a CC with 50% bone by volume would get an E of 14.5 GPa, because 0.5(22 – 7) + 7 = 14.5. Is that right? I think it would help your readers if you spell it out. Also, how was the range of 7 – 22 GPa selected?

L153: Are the percentages here the percentage of the space occupied by bone, or by air? i.e. for Enhydra lutris, are you deleting 16.5% of the solid material, or deleting 83.5% of the material so that only 16.5% remains? From the way it’s written, I thought it was the second one (only 16.5% remains), but later (L169) you refer to “16.5% volume deletion”. 16.5% deletion makes more sense (these animals aren’t made of air!) but this needs to be clearer, especially in the context of figuring out what the material properties are.

Paragraph at L213: I agree that the numbers for PC are closer to the CC than the numbers for MC, but saying that they “replicate” the peaks and valleys is a stretch. In both PC and MC, you have what looks like a smooth curve with the PC line being slightly more jagged. It certainly cannot replicate the extreme peaks of the CC. The only real difference between PC and MC is the VM stress values, and I think that if you compared the gradients of the lines of the PC and MC curves from point to point, you would find them almost indistinguishable from one another.
This is unsurprising, because the peaks in the CC are almost certainly due to “hotspots” of stress developing around individual trabeculae (evident in Fig. 4). How do your transect lines match up with actual material in the CC models? Are they interrupted at all, especially in the lower bone volume cylinders? Are they passing directly through hotspots?
All of this make me wonder is transecting across the cylinder is the best way of measuring “equivalent” performance among your cylinders: although those hotspots are real (i.e. not artefacts of boundary conditions), they are still incredibly localized. In a whole bone model, you are unlikely to be interested in such localized effects and would probably care more about whether the overall deformations are correct. I wonder, if you were to look at something like the overall strain (by which I mean change in the height of the cylinder before and after compression), would this not be a better way of seeing which of the experimental cylinders match the controls? This will average out those hot spots and give you a better idea of how deformation is accommodated across the whole model. This would then allow you to comment on something else I’ve been wondering about: the arrangement of trabeculae within bones is presumably not random. If a fabric was present in the bones, it would be present in the CC models. Whether you’re loading them in a manner that matches how that fabric would be loaded in life is moot – if the fabric is important, I expect that it’d show up in the overall deformation, but not (or less) in the hotspots.
EDIT: you do acknowledge the measurement method later, in the discussion (L273), but I don’t think anything as complicated as plates is necessary. For a start, the plates could impose their own stiffness even with low Young’s modulus values. Overall deformation of cylinder length (and maybe width too) would be easy to measure as a starting proxy for relative performance, and I think it would be a useful addition to the paper. A good test of this would be in your repeatability study (Fig. 2) – does an alternative method mitigate the artefacts that distort the results from test IV?
EDIT 2: you also have mentioned the trabecular arrangements on L282. I got ahead of myself! If I were you, I’d move your paragraphs about these limitations forward in the discussion and end with the discussions of the useful advice (see next comment), because that’s the key contribution that this paper makes.

L255: This is a really good point. I wonder if you can highlight it more? I feel like you haven’t talked much about this combined effect aspect of the models, but the fact that porosity is more important than properties is arguably the key take-home message. This could be even more important in animals with lower trabecular volumes (e.g. birds and pterosaurs), or less important in marine mammals using ballast, and you could expand on this in the discussion. A further point (raised in L268 but could be expanded on) is that this potentially explains why so many FE models have been found to be too stiff when validated. From memory, most of them use “solid” approximations of trabecular bone with reduced stiffness, but hardly any account for the differences in material distribution that real trabecular bone produces.

Figs 3 and 4: The species here are organised alphabetically, but I think it would be more useful to arrange them in terms of bone volume, so that it will be easier to see how changing the bone volume alters the results. I wonder if it would also be worth plotting all of the CC models on one graph, and all of the PC models on another, again so the readers can more easily discern the effect of bone volume on the results. For instance, I am curious as to whether there is a linear relationship between percentage bone volume and performance.

Minor Edits
L73: “The pushing of the boundary for FEA and better modeling of bone structures have been continuous for the last decade or so to better understand skeletal form and function”
This sentence would be better if rearranged: “For the last decade or so, the boundaries of FEA have been pushed towards the better modeling of bone structures to better understand skeletal form and function”
L81: “When in the presence of this type of porous structures” Should be “this type of porous structure” or “these porous structures”
L264: I’m still not convinced that any of the experimental models capture the peaks of the controls, and think this phrasing should be reconsidered (in the results, too). Also, I think you meant to have a comma after the word “trajectory”
Fig 1 caption: should say “fixed nodes”
Fig. 3: Can you make the fonts bigger on the axes and legend? These need a lot of zooming on a computer, and will be tiny printed out. Maybe make the lines on the graphs thicker too, it’s not easy to discern the colors at this line weight. Also, the image in Fig 1 and the supplementary tables say that you took four transects per cylinder at 25% intervals around the circumference, yet each model only has one line of these graphs. Can you explain this?
Fig 4: Can you write which species and bone volumes the rows represent on to the figure?

·

Basic reporting

The basic reporting is sound, with some possible improvements to the writing.

Experimental design

The research conforms to the aims and scope of PeerJ. The questions and implications are quite carefully defined. The methods for the innovative 3D erosion are a little unclear.

Validity of the findings

The research has highly important recommendations about assigning material properties to cancellous bone. Bulk/regional assignment is simply inferior to replication of trabecular networks through the authors' methods, and the approach will be a huge benefit for accurate replication of structural response of cancellous bone. The authors do well with negative controls, and convincingly explain the study as exploratory rather than statistical. The conclusions are well-stated and circumscribed.

Additional comments

This is an exciting study that will increase the accuracy of FE simulations for cancellous bone, at least at smaller size scales. I'm not certain at what scales the method will become impractical beyond the size range of cores the authors took from mammalian jaw joints. However, the erosion method is clever and may give results for back-calculating stiffness of large regions of cancellous bone. The figures could use larger font sizes, but are otherwise clear.

The main improvements to the manuscript will be to language. Pay attention to suggestions for more active writing and conciseness. For example, the heading "Element Reduction Script Verification Analyses" is all nouns for one concept. A more engaging heading would be "Does random element reduction deliver precise results?”. Some other passages also have a surfeit of gerunds. The acronyms (CC, PC, and MC) are hardly necessary, except in figures and explained in captions. PeerJ readers will be eclectic, and spelling out the terms the first time in each paragraph, or omitting the acronyms from the text, will prevent initial frustration.

---

## Round 0.2 · Minor Revisions

Thanks so much for all the hard work you've put into amending this manuscript. I am satisfied that you have addressed all the comments raised by the reviewers. There are just now a few outstanding points that I have noticed, listed below. I've also noted a number of language errors on the attached PDF. I don't think these further comments should take very long to address, so I look forward to seeing a revised version in the near future.

1. On line 145 you state that the 'remaining boundary conditions for the MC group were set up as in the CC group'. This is confusing as it leads the reader to expect that the boundary conditions have been described earlier in the control cylinder section. Actually they are described later in the script analyses section and then again in the model simulation parameters section. To me it makes more sense to have the boundary conditions set out once in the model simulation parameters section and simply to refer to that in other sections. However, note that you haven't described the constraints in the model simulation parameters section - that only happens in the script analyses section.

2. Figure 3 and SI table S5 both seem to suggest that the PC+MC analysis was not carried out for the Enhydra cylinder. However, figure 5 clearly has a data point for the PC+MC line at 16.5% porosity. So was this analysis carried out or not? If it was, why is the data missing from figure 3 and table S5. If not, why not, and how was the data point in figure 5 generated?

3. Figure 6 is a key figure, but at the moment it seems a little scrappy. The cylinders are not aligned to the edge of the figure (they all lean to the right) and the labels underneath each cylinder are not consistently placed (some of them, notably 10GPa in cylinder L, overlap the cylinder). Also, you've labelled the cylinders A to O, but then don't use those labels in the caption. Either do a proper key (e.g. A-C, Mellivora cylinder; D-F, Enhydra cylinder; etc), or get rid of the lettering.

4. In general, the supplementary info is much better arranged, but there's one thing I'm still confused about. In the supplementary tables, there are two columns for each transect. I'm not sure what these are because they're not labelled. I assume the first column is the Von Mises stress, but what is the second? Also, you should use points rather than commas to separate the decimals.

---

## Round 0.3 · accepted · Accept

Thanks for making those last few revisions. I look forward to seeing this manuscript being published and hopefully the technique becoming widely used.